# Health Indicator Similarity Analysis-Based Adaptive Degradation Trend Detection for Bearing Time-to-Failure Prediction

**Zhipeng Chen** [1], **Haiping Zhu** [1,*], **Liangzhi Fan** [2,*] and **Zhiqiang Lu** [1]

[1]  School of Mechanical Science and Engineering, Huazhong University of Science and Technology, Wuhan 430074, China
[2]  School of Mechanical Engineering and Automation, Wuhan Textile University, Wuhan 430200, China
*  Correspondence: haipzhu@hust.edu.cn (H.Z.); lzfan@wtu.edu.cn (L.F.)

**Abstract:** Time-to-failure (TTF) prediction of bearings is vital to the prognostic and health management of rotating machines. Owing to the shifty degradation trends (DTs) of bearings, it is still difficult to obtain accurate TTF prognostic results. To solve this problem, this paper proposes an online, continuously updated TTF prognostic method based on health indicator (HI) similarity analysis and DT detection. First, multiple degradation features are extracted and fused to construct principal component HI by using dynamic principal component analysis. Next, exponential degradation models are fitted using the HI values for future state prediction. By regarding several HI values as a tested segment, the DT is detected by analyzing the similarity of the tested segment and the fitted curve. Finally, TTF is predicted by extrapolating the DT to hit the estimated failure threshold. Two case studies based on public bearing datasets demonstrate the superiority of the proposed approach over state-of-the-art methods.

**Keywords:** time-to-failure prediction; rolling element bearings; exponential degradation model; similarity

## 1. Introduction

Bearings are basic but critical components in rotating machinery. Their failure may be a major cause of machine breakdowns, which accounts for nearly half of machinery accidents [1]. These unscheduled shutdowns lead to vast amounts of losses of time and costs. To reduce these avoidable expenses, the technology of time-to-failure (TTF) prediction for bearings to ring alarms in advance for loss avoidance needs to be studied [2,3]. With the development of TTF technologies in recent decades, TTF prediction methods can mainly be divided into two categories: model-based and data-driven methods. These two kinds of methods have both advantages and shortcomings. Therefore, many researchers devoted themselves to the TTF prediction of bearings in two different ways.

Data-driven methods for bearing TTF prediction have experienced a boom recently due to the development of machine learning technologies [4,5]. These methods are simple and do not require prior and theoretical knowledge by making full use of large volumes of historical data to attempt to infer inner fault modes of bearings. Cheng et al. developed a multi-dimensional recurrent neural network to estimate the TTF of bearings under variable operation conditions [6]. Song et al. increased the efficiency of temporal convolution modules on TTF prediction by weighting the contribution of different sensor data through an attention mechanism [7]. To deeply exploit the potentialities of data features, Wang et al. learned degradation features from time-frequency representations of vibration data by using a three-dimensional CNN for TTF prediction [8]. Ding et al. combined deep transfer metric learning and kernel regression for the TTF prediction of bearings under different situations [9]. Data-driven approaches are useful when mechanical principles and degradation mechanisms are complex or vague. However, the prediction accuracy of

data-driven methods requires a high quantity and quality of training data, which is a big challenge for real applications.

Model-based methods try to build mathematical functions to fit and predict the degradation progressions of bearings. A well-designed health indicator (HI) that can effectively reflect the degradation processes of bearings is usually taken as an input for the mathematical function. Generally, multiple features, including time-domain and frequency-domain features [10]; features calculated from the signal decomposition algorithms such as empirical mode decomposition (EMD) and complete ensemble EMD (CEEMD) [11]; and features obtained by artificial neural networks [12] are extracted and selected to construct the HI. Then, statistical estimation algorithms such as Kalman filtering [13] and particle filtering [14] are employed for parameter updating. Liao et al. proposed a continuous Bayesian updating approach for the TTF prediction of bearings with a Pairs-Erdogan model [15]. Cheng et al. combined a double-Winer process model with the Monte Carlo algorithm to achieve degradation process modeling and reliability estimation of machinery [16]. Li et al. employed an exponential degradation model whose parameters are optimized by using the particle filtering algorithm to estimate the TTF of bearings [10]. Kundu et al. proposed a Weibull accelerated failure time regression model for TTF prognostic of ball bearings considering both operation condition parameters and condition monitoring signals [17]. Most of these methods work well and are rooted in a succession of three stages that consist of HI construction, degradation model fitting, and TTF prediction. However, there are still two problems that are neglected in these studies.

One problem is that random changes in current degradation trends (DTs) often decrease the accuracy of TTF prediction. Taking the most used exponential model as an example, researchers leveraged it to fit monitored data, and then future degradation status can be predicted for TTF prediction [18]. In this way, satisfactory results can be obtained if the HI trajectory or DT maintains an exponential-like degradation profile. Unfortunately, the current DT often shifts under various inner or outer influence factors, which leads to deviations between the current DT and the assumed evolution trajectory deduced from the fitted degradation model [19]. This phenomenon can be common and inevitable in real situations. As a result, the accuracy of TTF prediction may be decreased, and deviations between the true TTF and the estimated one may increase over time. Regarding this problem, we think the main reason for it is that past samples take a majority superiority during the parameter optimization of degradation models, and hence recent samples reflecting current DTs receive little attention. In other words, the parameters of fitted degradation models can hardly be modified to adjust to the current DT since previously collected data take the larger part of training samples during optimization. To the best of our knowledge, the existing literature still lacks reliable solutions for this problem. Therefore, in this paper, we seek to fill this research gap by developing an adaptive DT detection method based on HI similarity analysis.

The other problem lies in the accurate estimation of failure thresholds (FTHs). While the literature has developed fine degradation models to predict future states of a tested unit and can give out TTF estimation by elaborating its degradation signals to hit the FTH, the employed FTH is often problematically assumed to be a deterministic value for all tested units [20]. To solve this problem, some scholars started to estimate FTH online for a particular unit in TTF prediction problems. Chehade et al. estimated FTH for each operating unit in real-time with fitted convex quadratic formulations based on degradation profiles of historical units [21]. Liu et al. employed statistical population information from historical populations to dynamically estimate the FTH [22]. The literature depends heavily on large amounts of historical data records with which population-wide characteristics can be captured. Nevertheless, this gives them the same dilemmas as the data-driven TTF prediction methods.

To address the problems mentioned above, a HI similarity analysis-based adaptive DT detection method was proposed for the TTF prediction of bearings. With this method, more reliable and accurate TTF predictions are expected to be obtained by online estimation

of the FTH of operating bearings and continuous updates of the TTF prediction. The steps are as follows. First, multiple representative degradation features, which are called compound features in this paper, are extracted from vibration signals and fused by dynamic principal component analysis (DPCA) to construct principal component HI. Then, the HI values are coupled with scaling parameters (SPs) for FTH estimation once bearings are identified as degraded by clustered support vector machine (CSVM). Finally, exponential degradation models are fitted using the HI values, and the DT is detected by analyzing the similarity of the degradation curve and truncated HI trajectory. The TTF of bearings can therefore be predicted by extrapolating the detected DT to meet the estimated FTH.

The main contributions of this paper are as follows:

(1) A bearing TTF prognostic approach is proposed to provide a continuously updated prediction of TTF by adaptive DT detection based on HI similarity analysis.
(2) The specific FTH for each tested bearing is dynamically estimated with online monitored data and a configured SP to improve the accuracy of TTF prediction.
(3) The DT of bearings is adaptively detected with the fitted degradation curve and truncated HI trajectory to address the issue of DT shifts.

The organization of the paper is summarized as follows. In Section 2, the system scheme of the proposed approach is introduced. Sections 3 and 4 provide two experimental studies on public bearing datasets to demonstrate the effectiveness and superiority of the proposed approach compared with other major approaches. Finally, the conclusion is drawn in Section 5.

## 2. System Scheme of Time-to-Failure Prediction of Bearings

The flowchart of the proposed prognostic approach consists of three parts, including HI construction, FTH estimation, and TTF prediction, as shown in Figure 1. Their detailed procedures are described as follows.

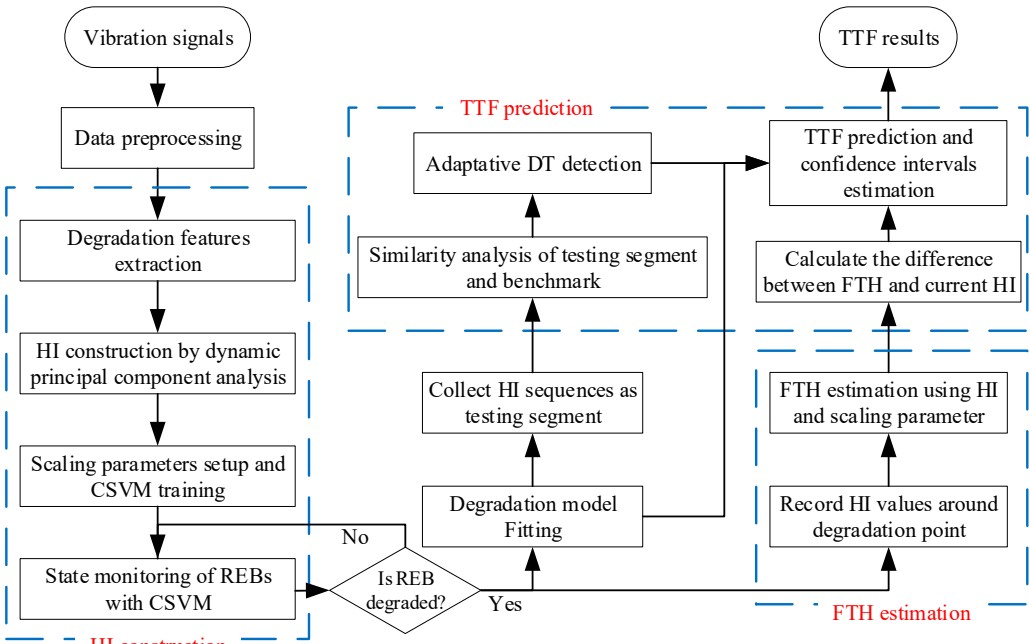

**Figure 1.** System scheme of time-to-failure prediction approach of bearings.

### 2.1. Health Indicator Construction by Fusing Compound Features

After raw vibration signals are collected from bearings through acceleration sensors, data preprocessing methods, including singular value elimination and Kalman filtering, are applied to eliminate noise and possible measurement error. After the preprocessing of the vibration signal is finished, compound features composed of time-domain features,

frequency-domain features, and intrinsic energy features are extracted for degradation evolution monitoring. Since the adopted features are classic and widely used, their mathematical equations are not listed, and the relative references where detailed explanations of these equations can be found are inserted.

First, five time-domain features, including root-mean-square-value (RMS), maximum absolute value (MA), kurtosis coefficient (KC), shape factor (SF), and clearance factor (CF), are calculated from the signals [23]. The RMS and MA indicate the amplitude and energy of the vibration signal over time, while the KC, SF, and CF describe the distribution over the time domain.

Second, to describe the changes in frequency components, the Fourier transform is utilized to convert the vibrational signal into frequency spectrums. Then, two statistical features, namely mean-square frequency (RMSF) and root variance frequency (RVF), are calculated to reflect the variation of the main frequency band and the dispersion of spectral energies [11].

Finally, the CEEMD is adopted to decompose vibration signals into 11 intrinsic mode functions (IMFs) [1]. The IMF indicates the natural oscillatory mode embedded in the signals and will change accordingly when the degradation of bearings occurs and produces resonance frequency components. Therefore, the intrinsic energy features of the IMFs are extracted as degradation indicators of bearings.

Once the extraction of compound features is completed, feature reduction is conducted to select reasonable degradation features that are well-correlated with the degradation process of bearings in consideration of monotonicity (Mon) and correlation (Corr). The mathematical expressions of these two metrics are written as

$$\begin{cases} Mon(F) = \frac{1}{K}|\sum_i \delta(f_{i+1} - f_i) - \sum_i (f_i - f_{i+1})| \\ Corr(F) = \frac{|N\sum_i if_i - N\sum_i f_i \sum_i i|}{\sqrt{[K\sum_i f_i^2 - (\sum_i f_i)^2][N\sum_i i^2 - (\sum_i i)^2]}} \end{cases} \tag{1}$$

where $K$ is the number of sampling points, $f_i$ denotes the $i$th feature value, and $\delta(\cdot)$ indicates the simple unit step function. The $F$ represents one of the extracted compound features.

For a comprehensive consideration, the average of the metrics is defined as feature selection criteria. The formula is

$$Score(F) = (Mon(F) + Corr(F))/2 \tag{2}$$

It can be found from the equation that the features with high scores should be selected.

These selected features will be further fused to construct the HI of bearings by using DPCA. This is an improved version of the principal component analysis (PCA) method, which can capture the most dynamic variations of vibration signals. The procedure of feature fusion by DPCA can be described as follows.

(1) Let the matrix $SF$ represents the selected $m$-dimensional features of bearings, which are defined as

$$SF = (F_1, F_2, \ldots, F_m) \tag{3}$$

where $SF_i = (f_{i1}, f_{i2}, \ldots, f_{im})$ denotes the $i$th normalized feature vector containing $m$ computed values.

(2) The score matrix $Sm$ is computed by transforming the matrix $SF$ according to the DPCA method, the form of which is

$$Sm = E \times SF \tag{4}$$

where $E = (E_1, E_2, \ldots, E_n)$ is the transform matrix that consists of eigenvectors $E_i = (e_{i1}, e_{i2}, \ldots, e_{im})^T$. The eigenvectors are related to the largest eigenvalues $\lambda_i$ corresponding to the correlation matrix of $SF$. The projected vectors in $S$ are the principal components (PCs) that are used for HI construction.

(3) The HI of bearings is constructed as follows:

$$HI_i = \sum_{j=1}^{m} \lambda_{0j} \lambda_j \sqrt{(Sm_{ij} - \mu)^T C^{-1}(Sm_{ij} - \mu)} \tag{5}$$

where $HI_i$ is the HI value of $i$th sampling, and $S_{*j}$ denotes the first PCs of the current sample. $\mu$ and $C$ are the learned mean and covariance of PCs decomposed from compound features of training bearings, respectively.

### 2.2. Failure Threshold Estimation Based on the Online HI Values

Healthy bearings always start with a stochastic stationary process, while the monitored vibration signals and their HI can also keep a relatively stable level. Once the HI values reach a particular threshold, it indicates that the bearings deteriorate, since the HI reflects the degradation progression of bearings. This time point is named a degradation point (DP). Due to the assumption that the degradation profiles of bearings are expected to be similar if they are made with the same processes, we suppose that the proportion of HI values at DP to the FTH of bearings is the same and is subject to a random variation [24]. Based on this assumption, the FTH of particular units can be estimated based on the online HI values of bearings.

To complete the estimation of FTH, a scaled HI-based method is proposed in this section. Specifically, all HI curves of historical bearings are built and normalized first. Next, a statistical SP is computed for the same type of bearings through the average of the ratios of HI values at DP to the last HI values before bearings are judged to have failed. After the SP is obtained, the life phase of bearings is divided into two stages, comprising the health stage and the degraded stage. Additionally, historical data of bearings are classified according to the stages to train a CSVM model for the health monitoring of bearings. Then, the CSVM is employed to achieve health status identification in this paper because of its excellent performance [25]. Finally, the FTH can be dynamically estimated when a tested unit is ready. The compound features are extracted from the monitored signal of the unit and inputted into the CSVM model for health status identification. Once the DP is detected, the HI value at the time instance is obtained and processed with the SP for the FTH estimation.

The diagnostic process of bearings using the CSVM can be described as follows. Let $Ds = \{x_1, x_2, \ldots, x_k\}$ denote the dataset that contains $k$ samples, where $x_k \in R^{m \times 1}$. Before training the CSVM, they are clustered into $C$ clusters $\{C_1, C_2, \ldots, C_c\}$ with the k-means algorithm. Within each cluster containing $C_l$ samples $(x_i^l, y_i^l)$, a linear SVM classifier $h_l(x)$ is built, and $1 \leq l \leq C$. The $h_l(x)$ is defined as

$$h_l(x) = \omega_l^T x \tag{6}$$

where $\omega_l$ is a weight vector, and the bias term $b$ is appended with an additional dimension. Therefore, the final classifier of CSVM is defined as

$$h(x) = \sum_l h_l(x) 1(x \in C_l) \tag{7}$$

where $1(\cdot)$ represents an indicator function. Finally, the objective function of CSVM is

$$\arg\min_{\omega, \omega_l, \xi_i^l} \frac{\lambda}{2} \|w\|^2 + \frac{1}{2} \sum_{l=1}^{C} \|\omega_l - w\|^2 + C \sum_{l=1}^{C} \sum_{i=1}^{n_l} \xi_i^l$$
$$s.t. \ y_i^l \omega_l^T x_i^l \geq 1 - \xi_i^l, \ i = 1, 2, \ldots n_l, \ \forall l \tag{8}$$

where $\xi_i^l$ are slack variables, $w$ is a global reference weight vector, and $\frac{1}{2} \sum_{l=1}^{C} \|\omega_l - w\|^2$ denotes global regularization. It should be noted that the global reference weight vector $w$ of CSVM is aligned with $\omega_l$ in each local linear SVM. In this way, information from other clusters can be shared to avoid over-fitting in each cluster. $\lambda$ is the weight of $w$, and $\xi_i^l$ denote slack variables.

After the training of CSVM on historical data is completed, it is employed to monitor the online bearings. Once a bearing is classified into the degraded, its HI is constructed, and the HI value at this time instance is obtained and named $HI_{dp}$. Subsequently, the FTH of the unit can be obtained by

$$FTH = (\sum_{i=-\tau}^{\tau} \frac{sw_i \times \Delta HI_{0,i+dp}}{\sum_i sw_i})/Sp + HI_{dp} \tag{9}$$

$$sw_i = \exp(-\frac{i^2}{2\tau^2}) \tag{10}$$

where $\Delta HI_{0,dp}$ represents the variation of HI from 0 to $dp$. Furthermore, $sw_i$ is introduced as sample weights for the $i$th sample to take into consideration sample priority.

### 2.3. Degradation Trend Detection Using Similarity Analysis for TTF Prediction

The TTF prediction is triggered and implemented when the bearings are identified to be degraded. To describe degradation progression, the exponential degradation model is employed because it can not only capture the degraded trend of bearings but also is easy to be fitted with fewer samples [26]. Its mathematical expression is as follows.

$$H(t) = a\exp(bt) + c \tag{11}$$

where $t$ indicates time, and $a$, $b$, $c$ are parameters that need to be determined by the least square algorithm [27].

Once the degradation model is successively fitted, generally, the TTF of bearings can be deduced by extrapolating the exponential degradation curve to hit the FTH. However, this prognostic result can be unsatisfactory sometimes due to the shift of the current DT. Consequently, the TTF prediction with the original fitted degradation model will be inaccurate before considering this situation. To address the problem, the method of HI similarity analysis with Fréchet distance is proposed to detect the DT of bearings adaptively [28]. The Fréchet distance has a great advantage of similarity measurement on time series curves because it takes the location and ordering of the points along the curves into account.

To analyse the similarity, the last $q$ HI values of bearings are cut off as a tested segment (Ts) to compare with the fitted degradation curve (Fc) point-wise starting from DP. Specifically, let the $Ts_t$ represents the Ts at time $t$, and $Fc_i$ indicate $q$ predicted values of the Fc from $t + I - q$ to $t + I$, where $i \in N^+$. Then, the similarity between them can be computed as

$$\theta_i(Fc_i, Ts_t) = \inf_{\tau,\varphi,x\in[0,1]} \max D(Fc_i(\tau(x)), Ts_t(\varphi(x))) \tag{12}$$

where $\tau(x)$ and $\varphi(x)$ are two monotone increasing functions with $\tau : [0,1] \to Fc$ and $\varphi : [0,1] \to Ts$, and $D(\cdot)$ represents the Euclidean distance. It can be found that the smaller Fréchet distance means a higher similarity of the $Fc_i$ and $Ts_t$, which should receive more attention.

The DT at the selected time point $t + i$ represents the most appropriate one, which is named *mad*; therefore, the time interval (TI) between $t$ and failure time can be calculated like

$$TI_t = \inf\{r : H(mad + r) - H(mad) \geq \Delta HI_t | HIs\} \tag{13}$$

$$\Delta HI_t = FTH - HI_t \tag{14}$$

where $TI_t$ is the TI at $t$, $HIs$ represent all HI values obtained at $t$, and $H(mad + r)$ denotes the predictive HI at $mad + r$. In addition, the predictive HI values follow a t-distribution, and the confidence interval can be computed as follows:

$$H(r) \pm t_{\alpha/2} s \sqrt{\frac{1}{n} + \frac{(r - \bar{t})^2}{\sum_{t=1}^{n} (t - \bar{t})^2}} \tag{15}$$

$$s = \sqrt{\frac{\sum_{t=1}^{n} \left( HI(t) - \overline{HI} \right)^2}{n - 2}} \tag{16}$$

where $\alpha$ is the confidence, $n$ is the number of collected HI values, and the $\overline{HI}$ indicates their mean value.

Finally, at time $t$, the TTF of bearings is obtained by adding the $TI_t$ to $t$, which is expressed as

$$TTF_t = TI_t + t \tag{17}$$

A schematic diagram of the computational steps on a general example is presented in Figure 2. At the time $t = 485$ min, the degradation curve is first fitted with the monitored data. Then, the $Ts_t$ composed of the latest serval HI points slides along the fitted curve to calculate their similarity, which is shown as S1, S2, and S3 in the figure. Supposing S2 is the best, the corresponding point of the fitted curve is selected as the DT. Finally, the $TI_t$, as well as the TTF, can be predicted using Equation (11).

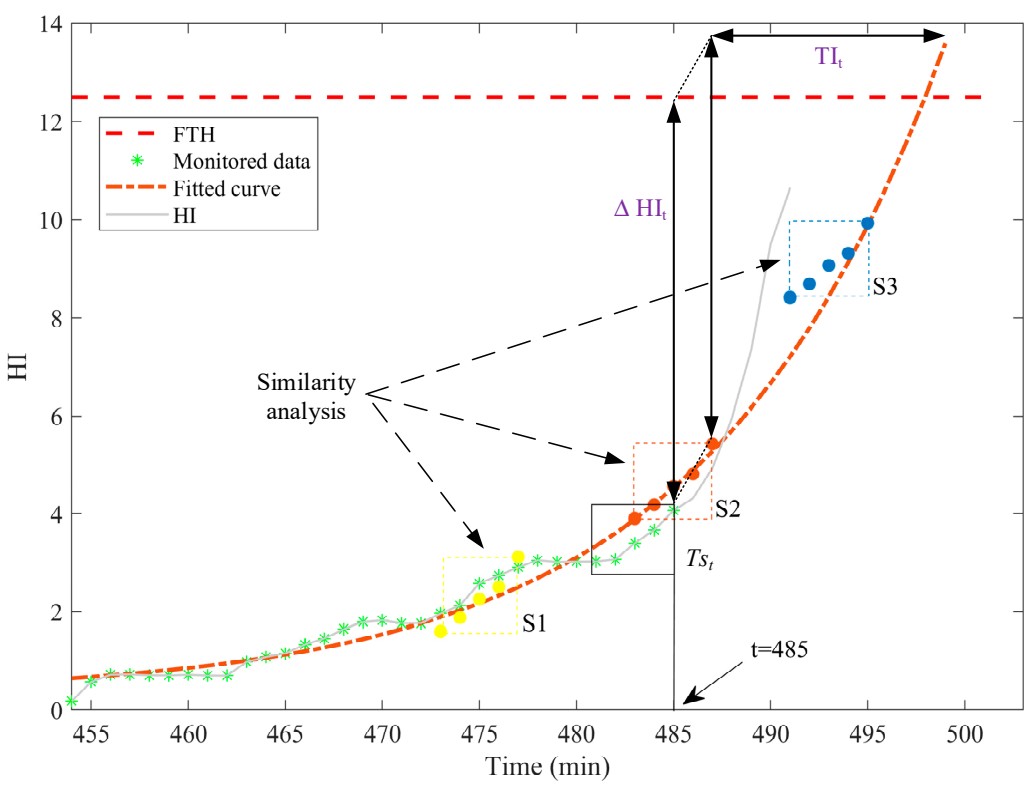

**Figure 2.** The schematic diagram of TTF prediction using the proposed method.

### 2.4. Performance Evaluation Metrics

To quantitatively evaluate the performance of the proposed approach, two evaluation metrics are employed. The first evaluates the error of TTF prognostics, which is given as

$$Er = \frac{\left| TTF - \overline{TTF} \right|}{TTF} \tag{18}$$

where $TTF$ is the actual TTF, and $\overline{TTF}$ represents the predicted TTF.

Since the error metric only checks the error of the last prediction result, another metric named CRA is adopted to assess the accuracy of the prognostic model comprehensively by accumulating the relative accuracy (RA) at each time point.

$$CRA = \sum_{i=1}^{n} nw_i RA_i \tag{19}$$

$$RA_i = 1 - Er_i \tag{20}$$

where $nw_i$ is a normalized weight factor and $nw_i = i/\sum i$. The $RA_i$ and $Er_i$ denote the RA and error of predicted TTF at the time $i$, respectively. Thus, the best CRA of models is 1.

### 3. Case Study 1: XJUT-SY Bearing Dataset

#### 3.1. Data Description

The XJTU-SY dataset is used to verify the proposed method [24] and to conduct comprehensive comparisons with other major state-of-the-art approaches. The test platform used for run-to-failure experiments consists of three main subsystems, including the power subsystem, bearing subsystem, and measurement subsystem, which are depicted in Figure 3. Fifteen bearings whose types both are LDK UER204 are adopted in the test, and two PCB 352C33 accelerometers are fixed orthogonally to collect the horizontal and vertical vibration signals. To monitor the degradation progression of bearings, the sampling frequency is set as 25.6 kHz with a duration of 1.24 s every min.

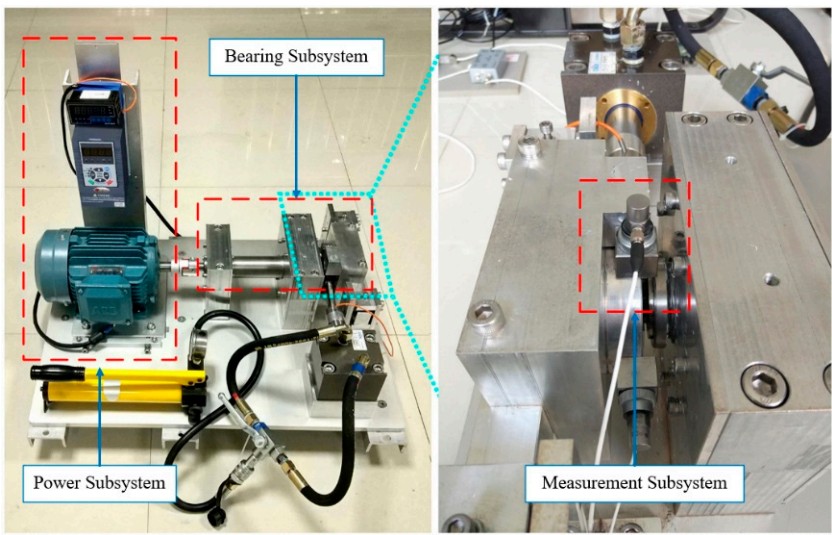

**Figure 3.** The experimental platform of XJUT-SY.

It should be noted that the horizontal vibration signals of bearing 1_3, bearing 1_5, bearing 2_1, and bearing 3_1 are adopted for the TTF prognostics of bearings since they show good degradation behaviors.

#### 3.2. Data Preprocessing and HI Construction

According to Section 2.1, compound features are extracted and selected to develop effective HI. The scores of all eighteen extracted features are shown in Figure 4. Based on the scores, a total of 14 features whose scores are greater than the selection threshold are selected for the following procedures.

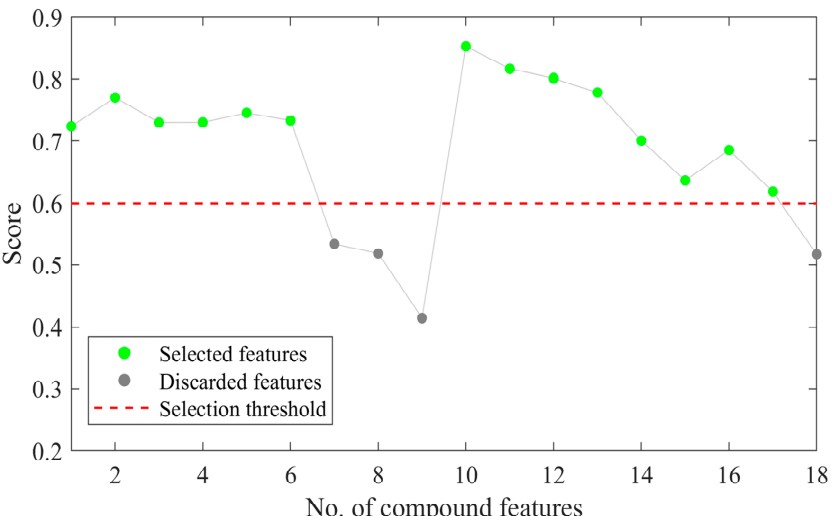

**Figure 4.** The scores of compound features of bearing 1_3.

The obtained HIs of all tested bearings are shown in Figure 5. The HIs contain a significant amount of effective degradation information after sequential procedures of compound features extraction, selection, and fusion, which makes them sensitive to the degradation phenomenon of bearings and appropriate for DP identification.

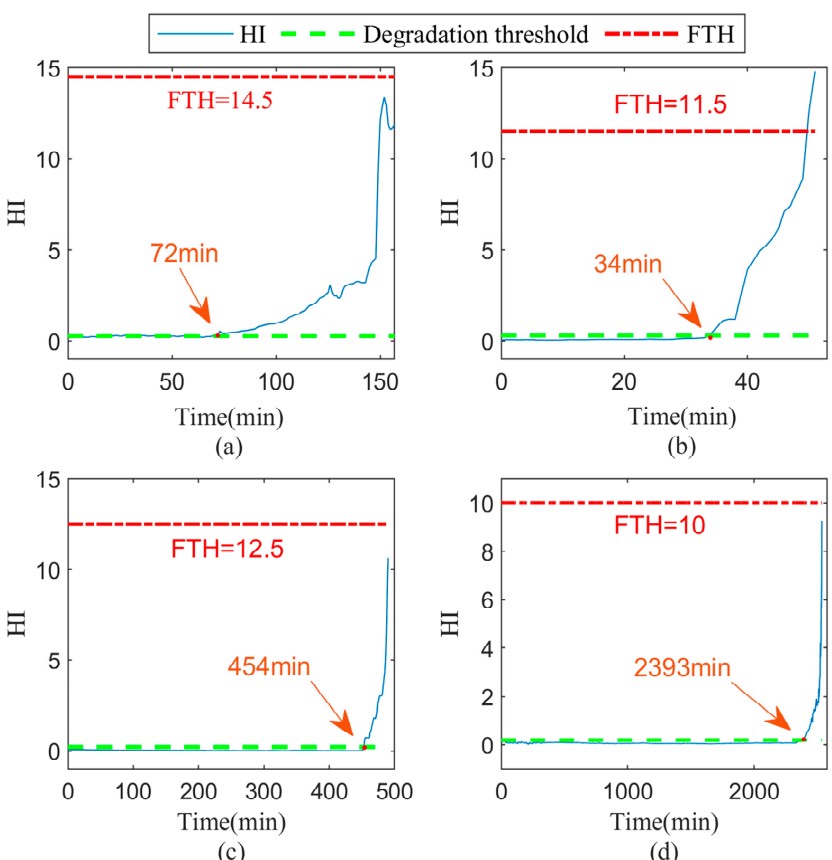

**Figure 5.** HIs for TTF prognostics of (**a**) bearing 1_3, (**b**) bearing 1_5, (**c**) bearing 2_1, and (**d**) bearing 3_1.

The SP is set to 0.02 in this study, and other bearings, except for the four tested bearings, are utilized to train the CSVM model. A grid search is employed to optimize the parameter

λ of the CSVM for high accuracy of DP detection, and the final value is determined as 0.1 after optimization. For the tested bearings, their DPs are identified to be 72 min, 34 min, 454 min, and 2393 min by the well-trained CSVM model, respectively.

Based on the detected DP as well as the configured SP, the FTH can be estimated with Equation (9). It can be found that the parameter τ controls the number of participants in FTH estimation and decides their weights, which has a significant impact on the accuracy of FTH prediction. Therefore, a trial-and-error method is adopted to determine a proper candidate, and the value of τ is finally decided as 3 after comparisons. The estimated FTHs for bearing 1_3, bearing 1_5, bearing 2_1, and bearing 3_1 are 14.5, 11.5, 12.5, and 10, respectively, as depicted in Figure 5.

### 3.3. TTF Prediction

When bearings are identified to be degraded, the TTF prognostic procedure will be triggered. The TTF prediction procedure of bearing 1_5 at time $t$ = 48 min is detailed in Figure 6a. At first, the available HI values, which are shown as green points, are employed to fit the exponential degradation model. Next, the last HI segment is truncated as $Ts_t$ for DT detection with the Fc, which is presented by a dotted orange line in Figure 6a. The $Ts_t$ circled by a yellow rectangle is used to calculate the similarity with the Fc point-wise for proper DT detection. After the *mad* with the best similarity, 0.8525, is selected, the $TI_t$ can be acquired with Equation (11). $\Delta HI_t$ in the equation is set as the difference value between the HI value at time $t$ and the estimated FTH, and the future HI values can be predicted by extrapolating the Fc. Consequently, the $TI_t$ is calculated as the time interval from *mad* to the time point when the predicted HI value increases by $\Delta HI_t$. Finally, the TTF of bearing 1_5 at time $t$ is deduced with Equation (15). Moreover, the 95% confidence interval of the TTF prediction result, which is depicted by solid orange lines, is also obtained with Equation (13) by calculating the upper and lower confidence limits of the predicted HI values.

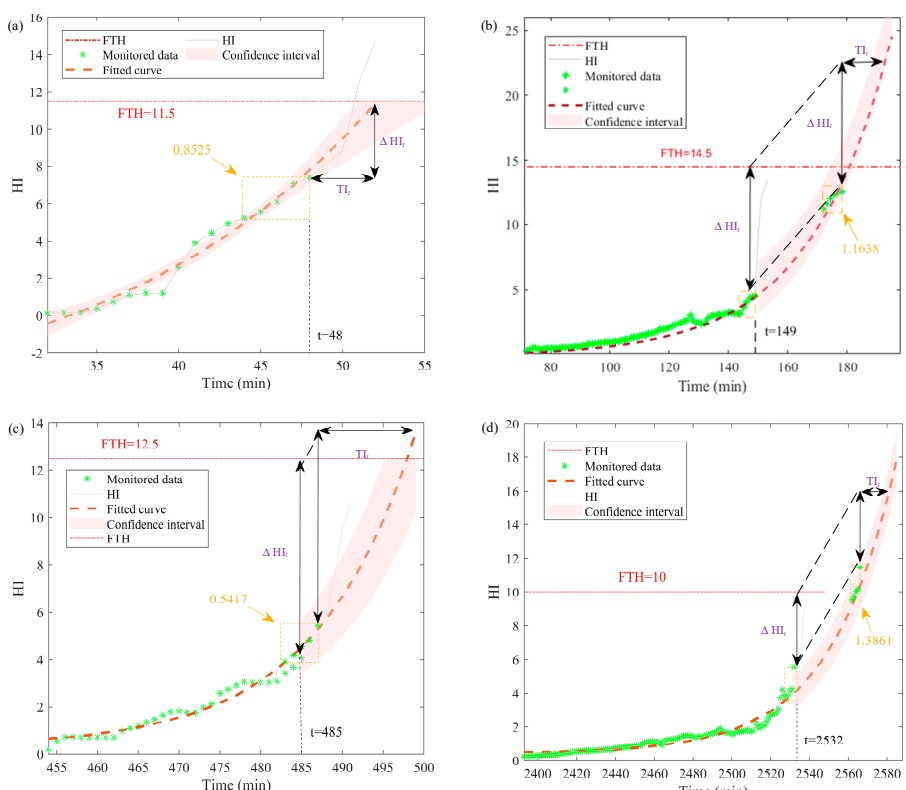

**Figure 6.** Degradation predictions of (**a**) bearing 1_5, (**b**) bearing 1_3, (**c**) bearing 2_1, and (**d**) bearing 3_1.

Further, the TTF prediction results of the remaining bearings at three different times are presented in Figure 6b–d, respectively. The predictive HI values are shown by dotted orange lines, and the *mad* at the inspection times are detected with the *Ts* that have the best similarity. It can be seen that the proposed approach can adaptively detect the proper DT at each time, which supports more accurate TTF prognostic results than the original prediction methods without DT detection. The TTF prediction results from the DP of all tested bearings are shown in Figure 7, where the 95% confidence interval of the predicted TTF is also presented. The TTF prognostics obtain dissatisfied results due to the lack of monitored data at the beginning but achieve increasing accuracy over time.

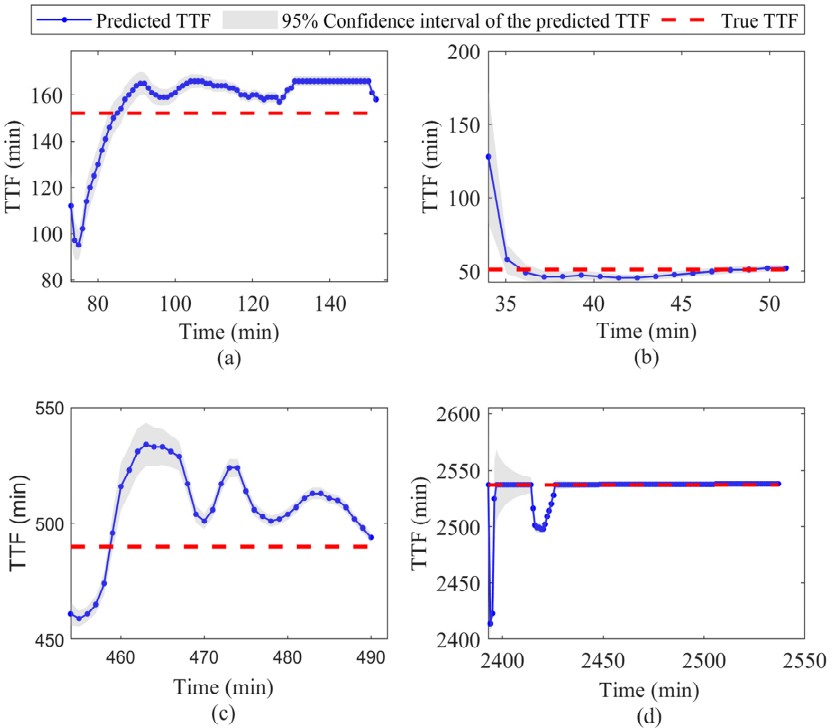

**Figure 7.** TTF prognostic results and its 95% confidence interval of (**a**) bearing 1_3, (**b**) bearing 1_5, (**c**) bearing 2_1, and (**d**) bearing 3_1.

### 3.4. Ablation Experiments

In the proposed approach, two technologies consisting of dynamic FTH estimation and adaptive DT detection are adopted to realize accurate TTF prognostics of bearings. To illustrate their advantages, a comprehensive comparison is conducted and shown in Figure 8. The origin method uses an exponential degradation model and a statistical mean FTH value from historical data to complete TTF prognostics. The other two methods replace the statistical FTH with the online estimated FTH for TTF predictions of bearings. Furthermore, the technology of DT detection is only employed in our proposed method. From the results in Figure 8, it can be found that the proposed approach has the best performance among all the three methods. Specifically, the TTF prognostics results predicted by the proposed approach not only achieve accuracy at the end of the time of bearings but can also quickly converge to actual TTFs after the start of the TTF prediction procedure. This is because the proposed method has the advantage of both FTH estimation and adaptive DT detection, which helps it to fit the degradation model with monitored data as well as identify DT via similarity analysis. Moreover, the method adopting the estimated FTH behaves better than the original one. This demonstrates the effectiveness of the dynamically estimated FTH on the improvement in TTF prediction accuracy.

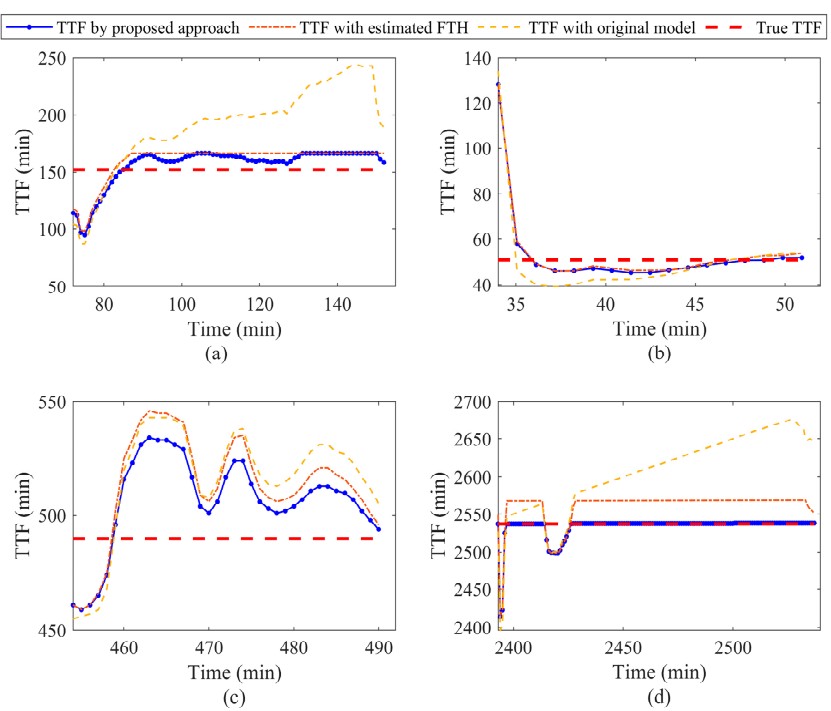

**Figure 8.** TTF prognostic results of the tested bearings (**a**) bearing 1_3, (**b**) bearing 1_5, (**c**) bearing 2_1, and (**d**) bearing 3_1 using an original method (without FTH estimation and DT detection), the method with FTH estimation, and the proposed method.

### 3.5. Comparisons with Other Approaches

To further demonstrate the superiority of the proposed approach, similar experiments with other major approaches, including long short-term memory (LSTM), support vector regression (SVR), and gaussian regression (GR), are also conducted. The parameter settings of these approaches are summarized as follows. In the LSTM network, 2 LSTM layers with neuron numbers of 24 and 16 are employed to handle inputted sensory data, and a fully connected layer is followed for TTF prediction. Typically, the Adam algorithm is utilized for network parameters optimization. The Gaussian kernel is employed in the SVR model. Additionally, the GR utilizes the RBF kernel to complete the prognostic task. After all the models are well-trained on historical data, the TTF prediction results of tested bearings are shown in Figure 9. It can be seen that the proposed approach gives more satisfying performances over other methods in terms of accuracy and convergence speed to the actual TTF. This can be mainly contributed to the ability of adaptive DT detection of the proposed approach with HI similarity analysis and the fitted Fc. Moreover, the numerical evaluation metrics are reported in Table 1, and the results from popular literature are also appended to the bottom of the table, where N/A represents that the information is not available. It can be observed that many models demonstrate their advantages, and the proposed approach obtains the smallest prediction errors and highest CRA scores. This means that the proposed TTF prognostic approach is superior to the compared models, and demonstrates the effectiveness of our method in the TTF prediction of bearings. It should be noted that the final 5% of all the sample cycles after the detected DP are collected for CRA calculation since the TTF prediction close to the end of the degradation progression is more important because most maintenance decisions are made at this phase.

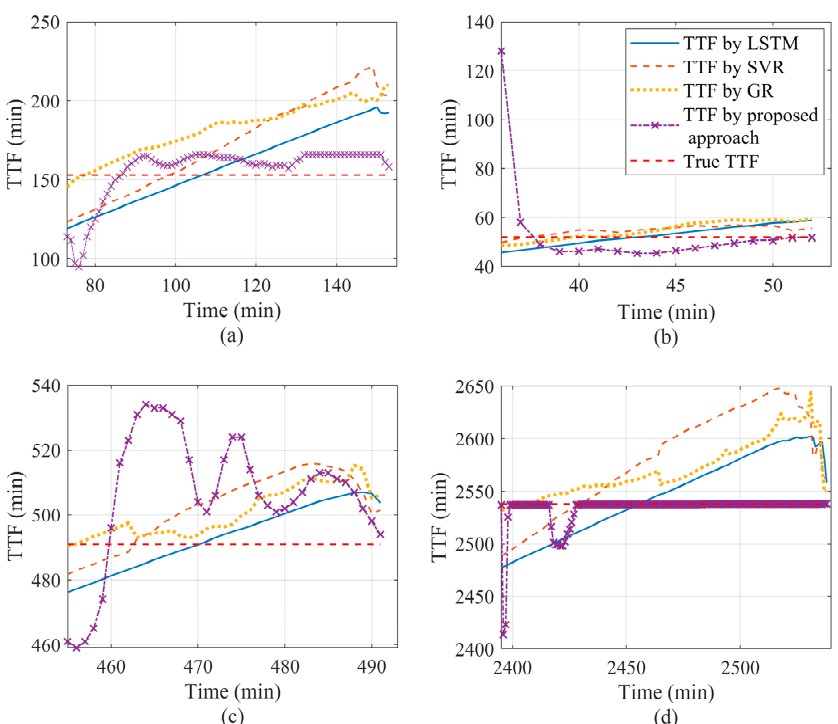

**Figure 9.** TTF prediction results of different approaches for (**a**) bearing 1_3, (**b**) bearing 1_5, (**c**) bearing 2_1, and (**d**) bearing 3_1.

**Table 1.** Performance evaluation for all approaches using two metrics.

| Method | Errors (%) | | | | CRA | | | |
|---|---|---|---|---|---|---|---|---|
| | Bearing 1_3 | Bearing 1_5 | Bearing 2_1 | Bearing 3_1 | Bearing 1_3 | Bearing 1_5 | Bearing 2_1 | Bearing 3_1 |
| Proposed approach | 3.27 | 0.11 | 0.61 | 0 | 0.9091 | 0.9827 | 0.9803 | 0.9987 |
| LSTM | 25.88 | 13 | 2.59 | 0.79 | 0.7162 | 0.8604 | 0.9651 | 0.9771 |
| SVR | 33.63 | 7.27 | 2.14 | 1.02 | 0.6148 | 0.9258 | 0.9736 | 0.9806 |
| GR | 37.25 | 14.05 | 3.09 | 0.49 | 0.6444 | 0.8567 | 0.9623 | 0.9768 |
| Wang et al. [26] | N/A | N/A | N/A | N/A | 0.8482 | 0.7878 | 0.8621 | 0.8942 |
| Sun et al. [29] | 7.01 | N/A | 2.59 | N/A | 0.53.6 | N/A | 0.5721 | N/A |
| Chang et al. [30] | N/A | 2.74 | N/A | N/A | N/A | 0.955 | N/A | N/A |
| Lin et al. [31] | N/A | N/A | 3.48 | 0 | N/A | N/A | 0.9727 | 0.9706 |

## 4. Case Study 2: FEMTO-ST Bearing Dataset

### 4.1. Data Description

The degradation data of bearings collected from the platform named PRONOSTIA were employed for another case study to validate the effectiveness of our proposed approach [32]. The overview of the PRONOSTIA platform is shown in Figure 10.

Four tested bearings, called bearing 1_1, bearing 1_2, bearing 2_1, and bearing 2_2, are selected as they show obvious degradation behaviors. Bearing 1_1 and bearing 1_2 performed run-to-failure experiments under a radial load of 4000 N at a speed of 1800 r/min, while the operation condition of bearing 2_1 and bearing 2_2 is an increased radial load of 4200 N with a reduced shaft speed of 1650 r/min. Their vibration signals were acquired by two accelerometers (DYTRAN3035B) mounted on the vertical and horizontal axes of the tested bearings. A total of 2560 samples were recorded with a sampling frequency of 25.6 kHz every 10 s. The signals from the horizontal accelerometers are adopted for the TTF prognostics of bearings in the following steps since they contain more useful information after comparisons.

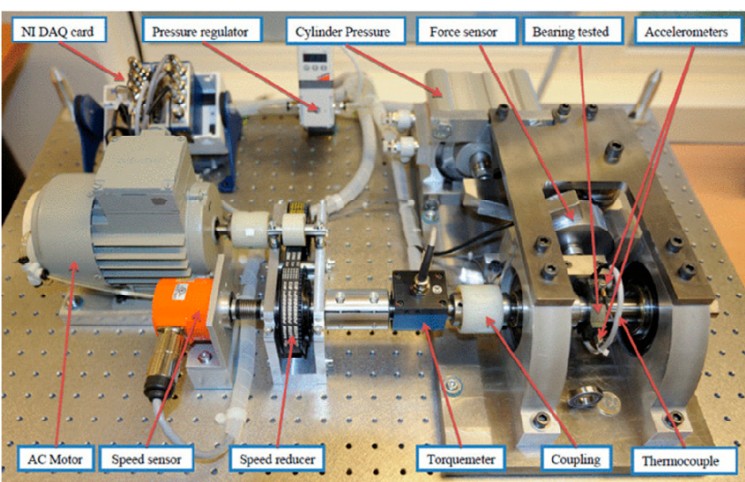

**Figure 10.** PRONOSTIA platform.

### 4.2. Data Preprocessing and HI Construction

The same steps, including noise elimination, feature extraction, and selection, are performed as in case study 1. Then, the HIs of the tested bearings are constructed by employing the DPCA and are presented in Figure 11. The DPs of bearing 1_1, bearing 1_2, bearing 2_1, and bearing 2_2 are identified to be 427.5 min, 126.2 min, 145.8 min, and 66.7 min, respectively. Their degradation thresholds are calculated and presented as dotted green lines when the SP is set as 0.2. Therefore, the FTHs for the bearings are estimated to be 14.3, 17.6, 17.7, and 15.2, as depicted in Figure 11.

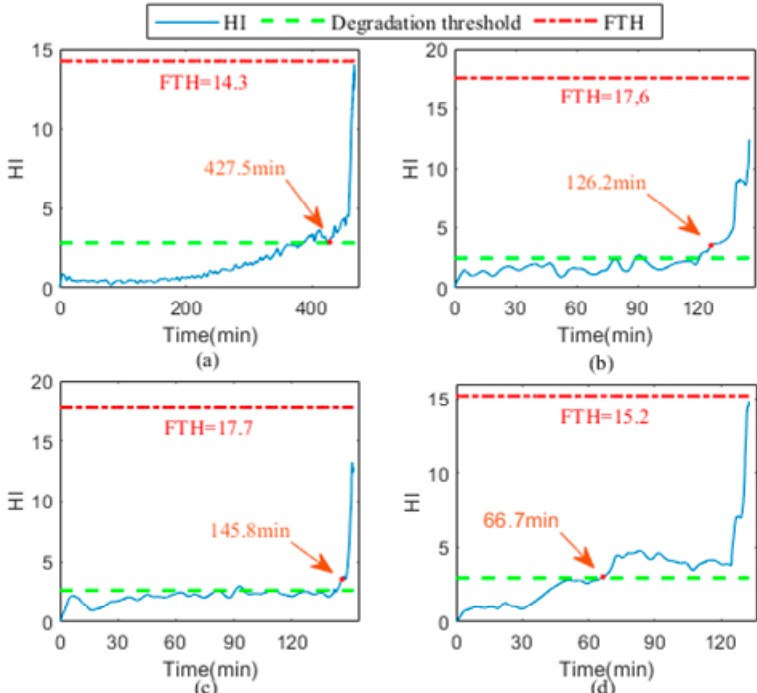

**Figure 11.** HIs for TTF prognostics of (**a**) bearing 1_1, (**b**) bearing 1_2, (**c**) bearing 2_1, and (**d**) bearing 2_2.

### 4.3. TTF Prediction

Several instances of the detailed description of the procedures for TTF prediction by using the proposed approach based on the tested bearings are presented in Figure 12. Taking bearing 2_1 as an example, the total prognostic process of TTF at the inspection time

of 150.5 min is shown in Figure 12c. First, the available HI values, which are plotted by green points, are adopted to fit the exponential degradation model marked with a dotted orange line. Next, the tested segment $Ts_t$ is acquired to detect proper DT. It can be found that the DT of the fitted degradation model at 150.5 min shows the best similarity with $Ts_t$ after exploration, which is 0.8735. This indicates that ordinary TTF prediction methods with only degradation model fitting can be a particular example of our proposed approach. Subsequently, future HI values can be predicted by extrapolating the fitted model from the detected DT, and the time interval $TI_t$ to the TTF can be calculated once the predictive HI values hit the estimated FTH from previous steps. Moreover, the 95% confidence interval of the predictive HI values, which is depicted by an orange area, is also obtained via Equation (15).

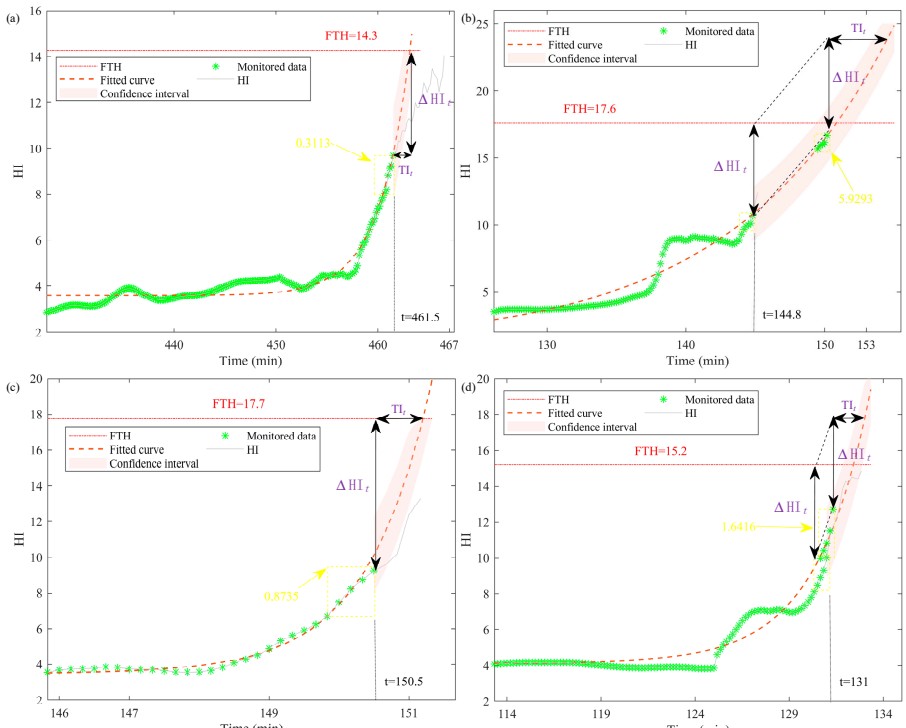

**Figure 12.** Degradation prediction of (**a**) bearing 1_1, (**b**) bearing 1_2, (**c**) bearing 2_1, and (**d**) bearing 2_2.

Further, the degradation prediction results of bearing 1_1, bearing 1_2, and bearing 2_2 at different inspection times are shown in Figure 12a,b,d, respectively. It can be seen that the successful DT detection by the proposed approach helps the fitted degradation model to address the problem of DT shifting. The groundwork for accurate TTF prognostics, as well as the increment in the feasibility of the proposed approach, is laid by this operation. The final TTF prediction results of the tested bearings are plotted in Figure 13 with a 95% confidence interval, which shows satisfactory performance and demonstrates the effectiveness of the proposed approach.

Three major methods, including the LSTM, SVR, and the original model (exponential model without DT detection and FTH estimation), are employed to compare with our proposed approach for further superiority validation. The parameter configurations of the LSTM and SVR are the same as those in case study 1. It can be concluded from the comparison results depicted in Figure 14 that the proposed approach achieves better TTF prediction accuracy than other competitors. What should be noted is that the predictive TTFs by the proposed approach show a fast convergence speed to the actual one as time goes on, though their deviation is huge at the beginning due to the lack of degradation data.

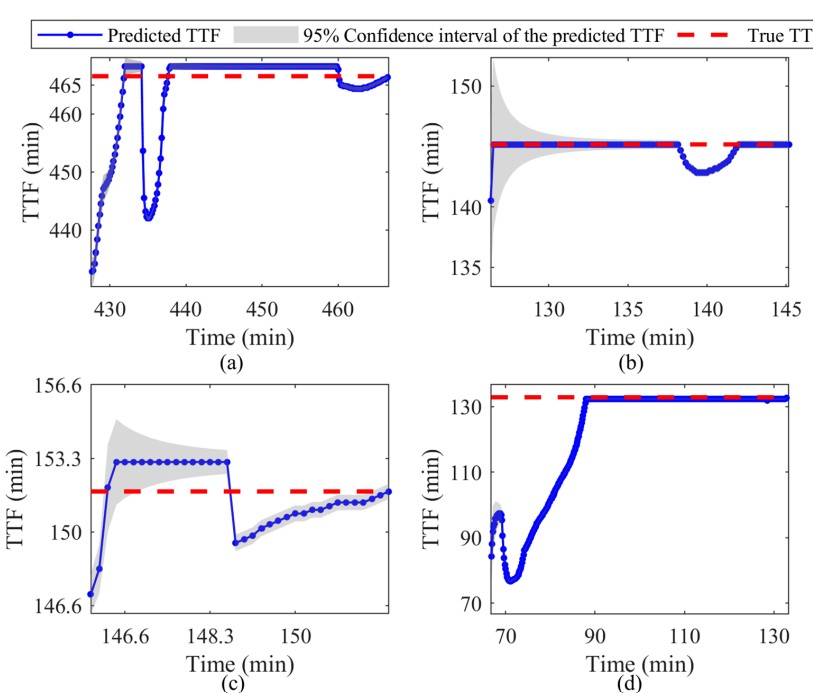

**Figure 13.** TTF prognostic results and the 95% confidence interval of (**a**) bearing 1_1, (**b**) bearing 1_2, (**c**) bearing 2_1, and (**d**) bearing 2_2.

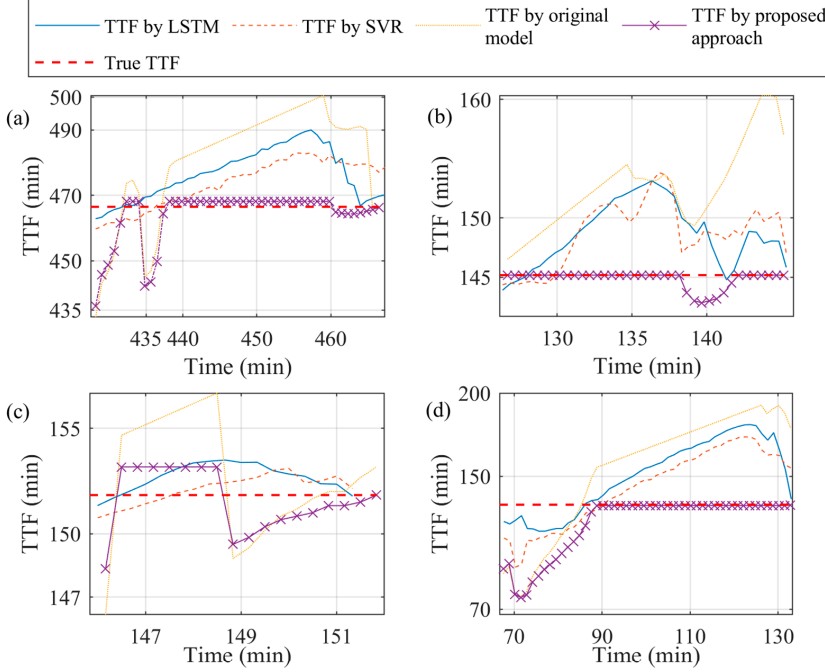

**Figure 14.** TTF prediction results of different approaches for (**a**) bearing 1_1, (**b**) bearing 1_2, (**c**) bearing 2_1, and (**d**) bearing 2_2.

In addition, the numerical metrics used in case study 1 are also employed for the quantitative evolution of the obtained results, which are summarized in Table 2. It can be identified that the proposed approach obtains the best grades among the outputs from comparative methods and other literature.

**Table 2.** Evaluation results of approaches considering two metrics.

| Method | Errors (%) | | | | CRA | | | |
|---|---|---|---|---|---|---|---|---|
| | Bearing 1_1 | Bearing 1_2 | Bearing 2_1 | Bearing 2_2 | Bearing 1_1 | Bearing 1_2 | Bearing 2_1 | Bearing 2_2 |
| Proposed approach | 0.04 | 0.0 | 0.0 | 0.13 | 0.9995 | 0.9954 | 1.0 | 0.9868 |
| LSTM | 0.6 | 0.34 | 0.3 | 2.44 | 0.9929 | 0.9767 | 0.9911 | 0.7643 |
| SVR | 2.4 | 1.18 | 0.54 | 16.4 | 0.974 | 0.9608 | 0.9885 | 0.7788 |
| Original model | 0.04 | 8.15 | 0.88 | 34.88 | 0.972 | 0.8912 | 0.9923 | 0.5458 |
| Wang et al. [26] | N/A | N/A | N/A | N/A | 0.9047 | 0.8546 | 0.8621 | 0.6521 |
| Wu et al. [11] | 0.02 | N/A | 0.22 | 0.37 | 0.98 | N/A | 0.78 | 0.63 |
| Cartella et al. [33] | 37.72 | 49.73 | 27.18 | 18.1 | N/A | N/A | N/A | N/A |
| Zhao et al. [34] | 13.9 | 65.55 | 47.2 | 17.3 | N/A | N/A | N/A | N/A |

*4.4. Discussion*

From the previous explanation of the TTF prediction process on inspection time points, it can be found that the proposed method can adjust to a proper *mad* to suit the current DT through HI similarity analysis. Therefore, a reasonable guess is that the proposed method can be applied to the TTF prediction of bearings under variable work conditions, since the changing load or rotation speed will impact the bearing degradation process, and this influence can be reflected by compound features as well as the HI curve. However, there is still a lack of proper bearing datasets for verification tests, and further efforts will seek proper situations to apply and verify the proposed method.

**5. Conclusions**

This paper proposed a novel TTF prediction approach for bearings using HI similarity analysis and adaptive DT detection. The approach mainly consists of three parts to obtain accurate TTF prognostic results. First, compound features are extracted from vibration signals and fused using the DPCA to construct effective HI for the degradation trajectory representation of bearings over time. Second, the CSVM model is trained to monitor the health state of bearings online. Once the bearings are identified to be degraded, their specific FTHs are dynamically estimated with monitored data, and the procedure of TTF prediction is trigged. Finally, the exponential degradation model is fitted to predict future states, and the HI similarity analysis-based DT detection method is adopted for TTF prediction. Unlike common methods, the proposed approach continuously updates the TTF to approximate the actual one, when more monitored data are acquired. The effectiveness and superiority of the proposed approach are verified with experimental demonstrations on bearing datasets.

In addition, this method can adaptively detect variational DTs in real working situations. Hence, the method has the potential to tackle the TTF prognostic problem of bearings under variable operating conditions. We will focus on this issue in future work.

**Author Contributions:** Conceptualization, Methodology, Software, Writing—Original draft preparation, Z.C.; Supervision, Funding acquisition, H.Z.; review and editing and validation, L.F. and Z.L. All authors have read and agreed to the published version of the manuscript.

**Funding:** This work was supported in part by the National Natural Science Foundation of China under Grant No. 52075202, and in part by the Key Research and Development Program of Hubei Province, China, under Grant No. 2021AAB001.

**Data Availability Statement:** Third-party data.

**Acknowledgments:** The authors express their gratitude to the editor and the anonymous reviewers for their valuable comments.

**Conflicts of Interest:** The authors declare no conflict of interest.

## Nomenclature

| | |
|---|---|
| $\{v_i\}_{i=1}^N$ | Raw vibration signal. |
| $f_i$ | Extracted compound features. |
| $HI_i$ | Constructed HI value. |
| $Sm$ | Score matrix computed in the DPCA algorithm. |
| $Ds$ | Input dataset for CSVM algorithm. |
| $C_l$ | The number of samples within the $i$th cluster. |
| $dp$ | Degradation time point. |
| $Sp$ | Scaling parameter. |
| $Ts$ | Tested segment. |
| $Fc$ | Fitted degradation curve. |
| $TI_t$ | Time interval at time $t$. |
| $mad$ | Detected the most appropriate degradation trend. |

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
