# Peer review of "Health Indicator Similarity Analysis-Based Adaptive Degradation Trend Detection for Bearing Time-to-Failure Prediction"

_electronics, doi:10.3390/electronics12071569_

Round 1

Reviewer 1 Report

The manuscript is clear and very interesting from a research point of view. I suggest to re-write the equations because are not in the correct format e.g. in line 207 the formula is up the text and not in line.

Reviewer 2 Report

The authors proposed a method for predicting the time to failures for bearings based on similarity analysis of health indicators and adaptive degradation trend detection. The test results were also provided to demonstrate the effectiveness of the proposed method. The paper is well-written.

The following changes are recommended:

(1). correction for typos. such as "de-tailed" in line 101.

(2). it is difficult to differentiate w and omega usages in lines 198-217.

(3). subscripts and lower case letters not easily to differentiate, such as sdt in Eq. (13) and line 245.

(4). It might be more appropriate to call Figure 2 as a schematic diagram instead of flowchart in line 260.   

(5). One question regarding feature selections: does the feature selections need to be updated as well? 

Reviewer 3 Report

The paper attempts to address the problem of detecting the bearing degradation state. Passing the paper to the Electronics section is infeasible. The paper addresses problems from machine construction and explanation.

There can be made several remarks about the presented problem.

1. Number of abbreviation is too large and make it extremally difficult to get through the text. Not all abbreviations are explained.

2. The formulas are put down in a strange way. There are many commonly used statistical factors (mean, standard deviation etc.) that are given in an expanded way that makes the formulas unnecessarily complex.

3. There is a missing explanation of the bearing degradation mechanism. It seems that authors attempt the blind search of features using extremely complex mathematical mechanisms.

4. There is a lack of discussion of measurement methodology. What about the changing load and rotation conditions? This will produce completely different results and feature distribution. The influence of other mechanical components on the detection process (gear operation, dynamically changing load).

5. The paper requires careful typesetting especially more attention should be paid to equation editing (see remarks on the manuscript)

I would like to recommend passing the paper to journals that focus on machine design or statistical methods in mechanics. The problems devoted to electronics were not exposed in the paper at all.

Please find an enclosed manuscript with remarks.

Round 2

Reviewer 3 Report

The paper after revisions still contains incorrect probabilistic equations (as shown in the previous review). Using the index sample for multiplying the sample is a severe flaw in the physical context. 

This paper cannot be accepted because the author's knowledge is insufficient in the area of the presented problem of probabilistic analysis.

The above serious flaws make the presented considerations at least questionable. There exists substantial evidence of a misunderstanding of the problem by authors.

I would like to encourage the authors to deeply revise the paper and resubmit after correcting all errors
